# Relationship between Kidney Stone Disease and Arterial Stiffness in a Taiwanese Population

**DOI:** 10.3390/jcm9061693

**Published:** 2020-06-02

**Authors:** Zih-Jie Sun, Hsuan-Jung Hsiao, Hsiang-Ju Cheng, Chieh-Ying Chou, Feng-Hwa Lu, Yi-Ching Yang, Jin-Shang Wu, Chih-Jen Chang

**Affiliations:** 1Department of Family Medicine, National Cheng Kung University Hospital, College of Medicine, National Cheng Kung University, No.138, Sheng Li Road, Tainan 70403, Taiwan; sunzihjie@gmail.com (Z.-J.S.); shiuanrong1313@hotmail.com (H.-J.H.); hsiangjucheng@gmail.com (H.-J.C.); jayin0421@hotmail.com (C.-Y.C.); fhlu@mail.ncku.edu.tw (F.-H.L.); yiching@mail.ncku.edu.tw (Y.-C.Y.); 2Department of Family Medicine, National Cheng Kung University Hospital, Dou-Liou Branch, College of Medicine, National Cheng Kung University, No.345, Zhuangjing Rd., Douliu City, Yunlin 64043, Taiwan; 3Department of Family Medicine, Ditmanson Medical Foundation Chia-Yi Christian Hospital, No.539, Zhongxiao Rd., East Dist., Chiayi 60002, Taiwan

**Keywords:** kidney stone, arterial stiffness, pulse wave velocity, age, gender

## Abstract

Previous studies examining the association between kidney stone disease (KSD) and arterial stiffness have been limited. Both age and gender have been found to have an impact on KSD, but their influence on the relationship between KSD and increased arterial stiffness is unclear. This study included 6694 subjects from October 2006 to August 2009. The diagnosis of kidney stone was based on the results of ultrasonographic examination. Increased arterial stiffness was defined as right-sided brachial-ankle pulse wave velocity (baPWV) ≥ 14 m/s. Associations between KSD and increased arterial stiffness were analyzed using multiple logistic regression models. KSD was positively related to increased arterial stiffness in both male and female groups (males: odds ratio [OR], 1.306; 95% confidence interval [CI], 1.035–1.649; females: OR, 1.585; 95% CI, 1.038–2.419) after adjusting for confounding factors. Subgroup analysis by age group (<50 and ≥50 years) showed a significant positive relationship only in the groups ≥ 50 years for both genders (males: OR, 1.546; 95% CI, 1.111–2.151; females: OR, 1.783; 95% CI, 1.042–3.054), but not in the groups < 50 years. In conclusion, KSD is associated with a higher risk of increased arterial stiffness in individuals aged ≥ 50 years, but not in those aged < 50 years for both genders.

## 1. Introduction

Increased arterial stiffness is a marker for increased risk of cardiovascular diseases arising from atherosclerosis, including coronary heart disease, heart failure, and stroke [1]. The clinical evaluation of arterial stiffness is commonly conducted by measuring pulse-wave velocity (PWV). Recent studies have demonstrated that brachial-ankle PWV (baPWV) is an independent predictor for morbidity and mortality associated with cardiovascular diseases [2,3,4]. Thus, the baPWV measurement is widely applied in the majority of Asian countries to evaluate arterial stiffness [5]. Moreover, old age, hypertension, and some risk factors of metabolic syndrome have been proven to be related to increased arterial stiffness [6].

Kidney stone disease (KSD) has traditionally been viewed as an isolated urinary system disorder. However, in recent decades, KSD has been found to be related to several systemic disorders, including obesity, metabolic syndrome, hypertension, diabetes, chronic kidney disease, and cardiovascular diseases [7,8,9,10,11,12]. However, data are insufficient regarding whether KSD is associated with arterial stiffness as a significant factor of cardiovascular disease. For these reasons, clarifying the association between KSD and arterial stiffness may have significant clinical implications for the prevention and diagnosis of early arterial stiffness as well as early atherosclerotic change by noticing the presence of KSD.

Currently, there have been only two studies on the association between KSD and arterial stiffness [13,14]. One study was based on a small population and demonstrated that people with calcium kidney stones had significantly higher values of carotid-radial PWV, carotid-femoral PWV, and augmentation index [13]. The other larger study conducted in a rural Chinese population indicated that KSD is significantly associated with increased baPWV [14]. Both age and gender have been found to have an impact on KSD [15,16], but their influence on the relationship between KSD and increased arterial stiffness is unknown. Therefore, we aimed to investigate the relationship between KSD and increased arterial stiffness by age group and gender.

## 2. Subjects and Methods

### 2.1. Study Population

This was a cross-sectional study for which the study population was drawn from subjects who had a health check-up at the Health Examination Center of the National Cheng Kung University Hospital from October 2006 to August 2009. They received different health check-up packages depending on their needs. Initially, we enrolled 7503 subjects aged ≥ 20 years who had undergone an abdominal sonography examination and a measurement of arterial stiffness. After excluding subjects with a history of stroke (*n* = 25) or heart disease (*n* = 152), either side of ankle-brachial index < 0.9 or > 1.3 (*n* = 177), abdominal sonography showing gouty nephropathy or uric acid deposition in the kidney (*n* = 43), and missing data (*n* = 467), a total of 6694 subjects were finally enrolled in this study for data analysis. This study was approved by the Ethical Committee for Human Research at the National Cheng Kung University Hospital in Taiwan (IRB number: B-ER-106-064). Data in this study were analyzed anonymously; thus, the informed consent of the subjects was not needed.

### 2.2. Data Collection

The demographic data, medical history, and lifestyle habits, including smoking, alcohol consumption, and exercise were collected using a structured questionnaire. Both smoking and alcohol consumption habits were classified into three groups as non-, ex-, and current-user. Current smoking was defined as smoking at least once per week for more than six months. Current alcohol consumption was defined as at least once drink per week for more than six months. Regular exercise was defined as exercise for at least 20 min each time and more than 3 times per week. 

Anthropometric measurements by well-trained nurses were performed using standard methods. Body mass index (BMI) was calculated as weight in kilograms divided by height in meters squared. Brachial systolic blood pressure (SBP) and diastolic blood pressure (DBP) were measured with an automatic blood pressure monitor, with the subjects in a supine position after resting for at least five minutes. Hypertension was diagnosed when subjects had a history of hypertension or SBP/DBP ≥ 140/90 mmHg. 

Blood samples were collected from all subjects after at least an 8 h overnight fast for measurement of fasting plasma glucose (FPG), glycated hemoglobin (HbA1c), total cholesterol, triglyceride, high-density lipoprotein cholesterol (HDL-C), creatinine, uric acid, calcium, and C-reactive protein (CRP) values. Except for subjects who were pregnant and those with a history of diabetes, the two-hour post-load glucose (2-h PG) level was checked after a 75-g oral glucose tolerance test. Diabetes mellitus was diagnosed when subjects had a history of diabetes, FPG ≥ 126 mg/dL, 2h-PG ≥ 200 mg/dL, or HbA1c ≥ 6.5%. The estimated glomerular filtration rate (eGFR) was calculated using the Modification of Diet in Renal Disease formula [17].

### 2.3. Ultrasound Imaging and KSD Diagnosis

All subjects received an abdominal ultrasound examination performed by experienced radiologists using a convex-type real-time electronic scanner (XarioSSA-660A, Toshiba, Tokyo, Japan) after at least an 8 h overnight fast. The diagnosis of kidney stone was based on ultrasonographic results, which showed echogenic foci with or without acoustic shadowing (small stones might not cast an acoustic shadow) in the renal pelvis or calices [18].

### 2.4. baPWV and Arterial Stiffness

We used the baPWV to evaluate arterial stiffness. The baPWV value was calculated as the pulse wave transmission distance divided by transmission time between the brachial and tibial arteries. It was measured automatically using a non-invasive vascular screening device (BP-203RPE II; Colin Medical Technology, Komaki, Japan) by wrapping pressure cuffs around each of the four extremities to measure the blood pressure and pulse waves of the brachial arteries of both arms and tibial arteries of both legs simultaneously after five minutes of bed rest. Increased arterial stiffness was defined as right-sided baPWV ≥ 14 m/s [19].

### 2.5. Statistical Analysis

The 17th version of SPSS software (Chicago, IL, USA) was used to perform the statistical analyses. Continuous variables were expressed as mean ± standard deviation, and categorical variables were presented as a number (percentage). In the univariate analyses, the Student’s *t*-test was used to compare continuous variables, and the Chi-square test was applied to compare categorical variables between subjects with and without increased arterial stiffness. The prevalence of increased arterial stiffness and KSD by age group in total population, males and females were compared using the Mantel–Haenszel Chi-square test for trend. In the multivariate analyses, the adjusted odds ratio (OR) and 95% confidence interval (CI) were calculated with binary logistic regression models to determine the relationship of KSD and other clinical variables with increased arterial stiffness. A *p* value < 0.05 was regarded as statistically significant.

## 3. Results

### 3.1. The Comparisons of Clinical Characteristics between Subjects with and without Increased Arterial Stiffness

Among the 6694 subjects, 2280 (34.1%) had increased arterial stiffness. Compared with subjects with normal arterial stiffness, subjects with increased arterial stiffness were older and male predominant, and they also had higher BMI, SBP, DBP, HbA1c, FPG, 2-h PG, total cholesterol, triglyceride, creatinine, uric acid, and calcium levels, but had lower HDL-C and eGFR values (Table 1). In addition, the percentages of the highest quartile of CRP, smoking, hypertension, diabetes, and KSD were found to be higher in the increased arterial stiffness group (Table 1). Moreover, the comparisons of the same variables between subjects with and without KSD are also shown in Table 2. The results were similar with that of the comparisons between subjects with and without increased arterial stiffness except for calcium and CRP ≥ 75th percentile (Table 2).

### 3.2. The Prevalence of Increased Arterial Stiffness and KSD by Age Group

The prevalence of both increased arterial stiffness and KSD showed increasing trends with increasing age in total population. The similar trends were also observed in both male and female groups. The prevalence of KSD peaked in the age group of 60–69 years in males. Males also had significantly higher prevalence of increased arterial stiffness and KSD compared to females (Figure 1).

### 3.3. The Risk of Increased Arterial Stiffness in Relation to KSD and Other Clinical Variables

After adjusting for age, gender, BMI, hypertension, diabetes, total cholesterol/HDL-C, triglyceride, eGFR, uric acid, calcium, highest quartile of CRP, smoking, alcohol consumption, and regular exercise, KSD was positively related to increased arterial stiffness (OR, 1.344; 95% CI, 1.095–1.649) (Table 3). The positive relationship remained the same in both the male and female groups (males: OR, 1.306; 95% CI, 1.035–1.649; females: OR1.585; 95% CI, 1.038–2.419) (Table 3). We further performed a subgroup analysis by age group (<50 and ≥50 years) (Table 4). We found the positive relationship was only significant in the groups ≥ 50 years for both genders (males: OR, 1.546; 95% CI, 1.111–2.151; females: OR, 1.783; 95% CI, 1.042–3.054), but was not significant in the groups < 50 years (males: OR, 1.106; 95% CI, 0.777–1.574; females: OR, 1.059; 95% CI, 0.496–2.259) (Table 4).

## 4. Discussion

This is the first study to investigate the impact of gender and age on the association between KSD and increased arterial stiffness. To the best of our knowledge, there have been only two studies examining the relationship between KSD and increased arterial stiffness, and gender and age were not examined as a factor in either study. These previous two studies demonstrated a positive relation between KSD and increased arterial stiffness [13,14]. However, one of the studies confined to calcium kidney stones was based on a small population (*n* = 84) [13]. Their study excluded post-menopausal women, so the study participants were younger than ours, and the unexposed group was composed of health professionals, not general population [13]. As for the other study, the subjects were confined to a rural population with a high prevalence of hypertension (70.7%) [14], which may not generalize the results. Another difference was that abnormal arterial stiffness was defined as any measurement of carotid-radial PWV, carotid-femoral PWV or augmentation index above the 90th percentile of the sample distribution in one study [13], a baPWV > 18 m/s in the other study [14], and a baPWV ≥ 14 m/s in our study. Both previous studies did not exclude people with an abnormal ankle-brachial index, which might have diminished the accuracy of the baPWV [20]. In addition, they did not exclude people with cardiovascular disease and did not adjust for possible confounding factors, such as age, hypertension, diabetes, lipid profile, renal function, uric acid, CRP, and a habit of regular exercise [13,14]. In our study, we found KSD to only be positively related to increased arterial stiffness for subjects aged ≥ 50 years, but not for those aged < 50 years, in both the male and female groups, after excluding subjects with abnormal ankle-brachial index, a history of stroke and heart disease, and adjusting for possible confounding factors. According to the results of the current study, the associated risk of increased arterial stiffness in people with KSD seems to be significantly influenced by age but not by gender.

The exact mechanism regarding the association between KSD and increased arterial stiffness is still unknown. First, although they share common risk factors such as aging, obesity, hypertension, diabetes, and dyslipidemia, the positive relationship between KSD and increased arterial stiffness remains the same after adjusting for the above risk factors. Second, chronic systemic inflammation may play a role in the common pathogenesis of KSD [21] and increased arterial stiffness [22]. CRP and interleukin-6 (IL-6) have been found to be common inflammatory markers for both KSD and increased arterial stiffness [21,22]. Although we adjusted for the highest quartile of CRP in the multivariate analysis, we cannot completely rule out contributions of inflammation other than CRP to both KSD and increased arterial stiffness. Third, vascular calcification may be another link between KSD and increased arterial stiffness. Because the active participation of the proteins and transcription factors found in bone formation are both involved in the early stages of kidney stone formation and vascular calcification [23], the extent of vascular calcification in people with KSD may be more obvious than those without KSD, and increased arterial stiffness seems to be the same. Additionally the physiological features of the vasculature may also play a role. Flow of blood from the descending vasa recta to the fenestrated ascending vasa recta decreases both vascular resistance and flow velocity in the ascending vasculature. The resulting turbulent blood flow increased the likelihood of injury at the renal papilla’s vascular environment, subsequently augmenting the formation of interstitial calcium phosphate deposits, known as Randall’s plaque (the inciting stone nidus) [23]. Vascular calcification may directly increase arterial stiffness. Alternatively, arterial stiffness may contribute to the development of calcification and focal plaque [24].

However, we found different results for the relationship between KSD and increased arterial stiffness in people aged ≥ 50 and <50 years. The aged artery is characterized by endothelial dysfunction, chronic inflammation, migration and proliferation of vascular smooth muscle cells, elastin fragmentation, extracellular matrix deposition, and matrix calcification/amyloidosis/glycation [25,26]. Levels of inflammatory mediators, especially IL-6 and CRP, increase with age in the absence of acute infection or other physiologic stress [27,28,29]. For example, Wei et al. found IL-6 concentration to be higher in males aged ≥ 55 years as compared to those < 55 years [27]. Wyczalkowska-Tomasiket al. confirmed higher levels of IL-6, CRP, and tumor necrosis factor receptor 1 (TNF-R1) in healthy individuals aged ≥ 65 years compared with those < 65 years [29]. In addition, calcification in the thoracic aorta free of intimal plaques was found to be most marked in the middle, elastin-rich layer of the media, with calcium content increased 30- to 40-fold from the age of 20 to 90 years [30]. One study examined 264 aortic specimens without intimal lesions and demonstrated that 44% and 100% presented with medial calcification in cases < 50 and ≥50 years of age, respectively [31]. Another study showed that the extent of vascular calcification increased exponentially with increasing age in both genders [32]. Thus, the degree of inflammation and vascular calcification increases with advancing age, especially after the age of 50 years. The influence of KSD on arterial stiffness, based on the points of inflammation and vascular calcification, may be more significant in people aged ≥ 50 years, but not in those aged < 50 years.

Uric acid was positively associated with increased arterial stiffness in the female group but not in the male group in this study. This result was similar to that of a Korean study, which found that elevated serum uric acid was independently associated with increased baPWV in healthy Korean women, but not in men [33]. Serum uric acid may promote vascular smooth muscle proliferation, and hyperuricemia may cause endothelial dysfunction [34,35]. The gender difference might be due to women being more susceptible to uric acid-associated vascular injury than men [36,37]. In the present study, calcium level was associated with elevated risk of increased arterial stiffness in the total, male, and young male groups. Madero et al. showed that serum calcium was not related to aortic pulse wave velocity in community-living elderly persons [38]. Ru et al. found serum calcium level to be related to baPWV in middle-aged and elderly Chinese subjects, especially in females and in subjects aged ≥ 65 [39]. Kimura, et al. demonstrated that longitudinal changes in serum calcium levels accelerated progression of arterial stiffness, especially in people aged ≥ 48 [40]. Further studies are needed for the inconsistent relationship between serum calcium and arterial stiffness although elevated serum calcium levels were found to induce the mineralization of vascular smooth muscle cells and then accelerate vascular calcification [23].

It is well known that aging and hypertension are factors that are strongly related to increased arterial stiffness, whereas the correlation between increased arterial stiffness and other established cardiovascular risk factors such as diabetes mellitus, dyslipidemia, renal dysfunction, BMI, and smoking have been shown to be less strong or insignificant [41,42,43]. The results in this study concurred with previous findings. Similar to arterial stiffness, the prevalence of KSD also showed a significant increasing trend with advancing age, but it declined in the age group ≥ 70 years of total subjects (Figure 1). The explanation for this may be that aged people have reduced urinary concentrating ability in response to vasopressin [44], which may associate with hypercalciuria, the most common metabolic abnormality in calcium stone formers [45]. Exercise has anti-oxidative and anti-inflammatory effects and can enhance the production of NO and reduce the concentrations of vasoconstrictors, so it has beneficial effect on vascular compliance and arterial stiffness [46,47,48]. Moreover, exercise intensity has been found to be typically higher in males than in females [49], and some studies have shown there to be no significant effect on arterial stiffness following exercise in elderly populations [50,51]. Therefore, the above finding was consistent with the results of this study suggesting that the beneficial effect of exercise on arterial stiffness was more significant in young males.

There are some limitations in this study. First, due to the cross-sectional design, we could not elucidate the causal relationship between KSD and arterial stiffness. Second, this study was based on a Taiwanese population; thus, caution should be taken while applying the results to other ethnic groups. Third, the composition of kidney stones is unknown; hence we could not further specify the association between different types of kidney stones and arterial stiffness. Fourth, bilateral or multiple stones were related to higher rates of metabolic abnormalities [52,53], but were not significantly associated with increased arterial stiffness in our further analyses. This may be because subjects with bilateral or multiple KSD were few and asymptomatic with less disease severity, resulting in non-significant results. Finally, dietary factors and medication effects were not considered due to a lack of relevant information.

In conclusion, this study demonstrated that KSD is associated with a higher risk of increased arterial stiffness in individuals aged ≥ 50 years, but not in those < 50 years for both genders. Risk of cardiovascular diseases may be evaluated in people aged ≥ 50 years with KSD. Further studies are warranted to investigate the underlying mechanism linking KSD and increased arterial stiffness by age group.

## Figures and Tables

**Figure 1 jcm-09-01693-f001:**
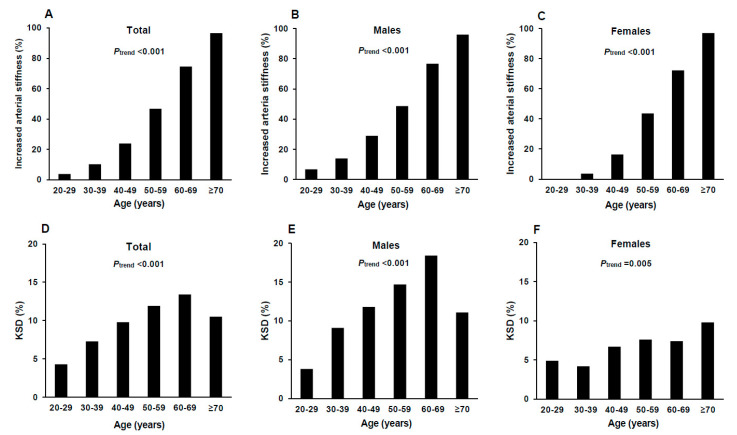
The prevalence of increased arterial stiffness and KSD by 10-year age group in total population (**A**,**D**), males (**B**,**E**) and females (**C**,**F**). KSD: kidney stone disease. The Mantel–Haenszel Chi-square test for trend was used to compare the prevalence of increased arterial stiffness and KSD among age groups.

**Table 1 jcm-09-01693-t001:** Comparisons of clinical characteristics between subjects with and without increased arterial stiffness.

	Increased Arterial Stiffness		
	No (*n* = 4414)	Yes (*n* = 2280)	*p* Value
Age, years	43.1 ± 1.0	56.1 ± 10.9	<0.001
Male gender	2546 (57.7)	1454 (63.8)	<0.001
BMI, kg/m^2^	23.8 ± 3.5	24.9 ± 3.4	<0.001
SBP, mmHg	112.7 ± 11.8	132.2 ± 16.0	<0.001
DBP, mmHg	66.9 ± 9.3	79.0 ± 10.4	<0.001
FPG, mg/dL	87.2 ± 15.1	94.1 ± 22.6	<0.001
2-h PG, mg/dL	109.1 ± 42.3	135.9 ± 59.3	<0.001
HbA1c, %	5.6 ± 0.6	5.9 ± 0.8	<0.001
Total cholesterol, mg/dL	193.5 ± 35.9	204.9 ± 37.8	<0.001
Triglyceride, mg/dL	119.1 ± 82.2	142.1 ± 95.1	<0.001
HDL-C, mg/dL	52.3 ± 14.4	50.1 ± 14.1	<0.001
Total cholesterol/HDL-C	4.0 ± 1.2	4.4 ± 1.3	<0.001
Creatinine, mg/dL	0.8 ± 0.2	0.9 ± 0.4	<0.001
eGFR, mL/min/1.73 m^2^	96.9 ± 17.3	89.8 ± 18.8	<0.001
Uric acid, mg/dL	5.7 ± 1.5	6.1 ± 1.5	<0.001
Calcium, mg/dL	9.2 ± 0.3	9.3 ± 0.4	<0.001
CRP ≥ 75th percentile	948 (21.5)	716 (31.4)	<0.001
Hypertension	215 (4.9)	912 (40.0)	<0.001
Diabetes	167 (3.8)	320 (14.0)	<0.001
KSD	353 (8.0)	307 (13.5)	<0.001
Smoking			0.04
Non	3741 (84.8)	1885 (82.7)	
Ex	264 (6.0)	171 (7.5)	
Current	409 (9.3)	224 (9.8)	
Alcohol consumption			0.68
Non	3817 (86.5)	1956 (85.8)	
Ex	113 (2.6)	65 (2.9)	
Current	484 (11.0)	259 (11.4)	
Regular exercise	340 (7.7)	161 (7.1)	0.35

Data are presented as mean ± SD or number (%). BMI: body mass index; SBP: systolic blood pressure; DBP: diastolic blood pressure; HbA1c: glycated hemoglobin; FPG: fasting plasma glucose; 2-h PG: two-hour post-load glucose; HDL-C: high-density lipoprotein cholesterol; eGFR: estimated glomerular filtration rate; CRP: C-reactive protein; KSD: kidney stone disease. The Student’s t-test for continuous variables or the Chi-square test for categorical variables.

**Table 2 jcm-09-01693-t002:** Comparisons of clinical characteristics between subjects with and without kidney stone disease (KSD).

	KSD		
	No (*n* = 6034)	Yes (*n* = 660)	*p* Value
Age, years	47.2 ± 12.0	50.4 ± 11.1	<0.001
Male gender	3517 (58.3)	483 (73.2)	<0.001
BMI, kg/m^2^	24.1 ± 3.5	24.6 ± 3.1	0.001
SBP, mmHg	118.9 ± 16.2	123.8 ± 16.3	<0.001
DBP, mmHg	70.6 ± 11.2	74.9 ± 11.4	<0.001
FPG, mg/dL	89.3 ± 18.1	91.6 ± 20.2	0.003
2-h PG, mg/dL	117.6 ± 49.9	123.7 ± 54.7	0.003
HbA1c, %	5.67 ± 0.7	5.74 ± 0.7	0.006
Total cholesterol, mg/dL	197.1 ± 37.2	200.6 ± 34.1	<0.001
Triglyceride, mg/dL	125.8 ± 86.3	137.5 ± 97.7	<0.001
HDL-C, mg/dL	51.7 ± 14.4	49.5 ± 13.6	<0.001
Total cholesterol/HDL-C	4.1 ± 1.3	4.3 ± 1.2	<0.001
Creatinine, mg/dL	0.86 ± 0.29	0.90 ± 0.19	0.002
eGFR, mL/min/1.73 m^2^	94.7 ± 18.2	92.4 ± 18.0	0.002
Uric acid, mg/dL	5.8 ± 1.5	6.1 ± 1.6	<0.001
Calcium, mg/dL	9.2 ± 0.4	9.2 ± 0.4	0.813
CRP ≥ 75th percentile	1484 (24.6)	180 (27.3)	0.131
Hypertension	979 (16.2)	148 (22.4)	<0.001
Diabetes	422 (7.0)	65 (9.8)	0.007
Increased arterial stiffness	1973 (32.7)	307 (46.5)	<0.001
Smoking			0.01
Non	5085 (84.3)	541 (82.0)	
Ex	374 (6.2)	61 (9.2)	
Current	575 (9.5)	58 (8.8)	
Alcohol consumption			0.218
Non	5214 (86.4)	1956 (84.7)	
Ex	154 (2.6)	65 (3.6)	
Current	666 (11.0)	259 (11.7)	
Regular exercise	448 (7.4)	53 (8.0)	0.574

Data are presented as mean ± SD or number (%). BMI: body mass index; SBP: systolic blood pressure; DBP: diastolic blood pressure; HbA1c: glycated hemoglobin; FPG: fasting plasma glucose; 2-h PG: two-hour post-load glucose; HDL-C: high-density lipoprotein cholesterol; eGFR: estimated glomerular filtration rate; CRP: C-reactive protein; KSD: kidney stone disease. The Student’s t-test for continuous variables or the Chi-square test for categorical variables.

**Table 3 jcm-09-01693-t003:** Multiple logistic regression analyses for the relationship of KSD and other clinical variables with increased arterial stiffness in both genders.

	Total (*n* = 6694)	Males (*n* = 4000)	Females (*n* = 2694)
	OR (95% CI)	OR (95% CI)	OR (95% CI)
Age, years	1.122 (1.114–1.131) ^c^	1.110 (1.099–1.120) ^c^	1.143 (1.127–1.159) ^c^
Male gender	1.145 (0.966–1.358)	-	-
BMI, kg/m^2^	1.006 (0.984–1.029)	0.992 (0.964–1.022)	1.014 (0.977–1.052)
Total cholesterol/HDL-C	1.069 (0.999–1.144)	1.045 (0.966–1.130)	1.134 (0.995–1.292)
Triglyceride, mg/dL	1.001 (1.000–1.002) ^b^	1.001 (1.000–1.002) ^a^	1.136 (0.993–1.301)
eGFR, mL/min/1.73 m^2^	1.004 (1.000–1.008) ^a^	1.005 (0.999–1.010)	1.003 (0.996–1.009)
Uric acid, mg/dL	1.079 (1.018–1.144) ^a^	1.035 (0.969–1.105)	1.164 (1.038–1.307) ^a^
Calcium, mg/dL	1.491 (1.231–1.806) ^c^	1.463 (1.147–1.868) ^b^	1.288 (0.933–1.779)
CRP ≥ 75th percentile	1.178 (1.009–1.374) ^a^	1.126 (0.936–1.354)	1.265 (0.954–1.678)
Hypertension, yes vs. no	7.936 (6.595–9.549) ^c^	7.308 (5.902–9.049) ^c^	10.621 (7.230–15.601) ^c^
Diabetes, yes vs. no	1.546 (1.211–1.973) ^c^	1.698 (1.260–2.289) ^b^	1.161 (0.756–1.783)
KSD, yes vs. no	1.344 (1.095–1.649) ^b^	1.306 (1.035–1.649) ^a^	1.585 (1.038–2.419) ^a^
Smoking			
Ex vs. non	0.991 (0.747–1.316)	1.032 (0.779–1.368)	0.232 (0.027–1.965)
Current vs. non	1.498 (1.176–1.907) ^b^	1.441 (1.126–1.844) ^b^	1.459 (0.548–3.886)
Alcohol consumption			
Ex vs. non	0.652 (0.427–0.995) ^a^	0.705 (0.458–1.086)	0.754 (0.174–3.261)
Current vs. non	0.847 (0.673–1.065)	0.898 (0.710–1.137)	0.590 (0.250–1.391)
Regular exercise, yes vs. no	0.705 (0.552–0.899) ^b^	0.605 (0.454–0.807) ^b^	1.084 (0.690–1.705)

KSD: kidney stone disease; OR: odds ratio; CI: confidence interval; BMI: body mass index; HDL-C: high-density lipoprotein cholesterol; eGFR: estimated glomerular filtration rate; CRP: C-reactive protein. ^a^ Statistically significant, *p* < 0.05. ^b^ Statistically significant, *p* < 0.01. ^c^ Statistically significant, *p* < 0.001.

**Table 4 jcm-09-01693-t004:** Multiple logistic regression analyses for the relationship of KSD and other clinical variables with increased arterial stiffness by age and gender groups.

	Males (*n* = 4000)		Females (*n* = 2694)	
	<50 y/o (*n* = 2367)	≧50 y/o (*n* = 1633)	<50 y/o (*n* = 1506)	≧50 y/o (*n* = 1188)
	OR (95% CI)	OR (95% CI)	OR (95% CI)	OR (95% CI)
Age, year	1.100 (1.078–1.122) ^c^	1.156 (1.129–1.184) ^c^	1.180 (1.130–1.231) ^c^	1.148 (1.119–1.178) ^c^
BMI, kg/m^2^	1.018 (0.980–1.057)	0.963 (0.920–1.008)	1.024 (0.959–1.093)	1.003 (0.960–1.049)
Total cholesterol/HDL-C	1.002 (0.896–1.119)	1.133 (1.007–1.274) ^a^	1.191 (0.915–1.551)	1.099 (0.938–1.287)
Triglyceride, mg/dL	1.001 (1.000–1.003) ^a^	1.002 (1.000–1.003) ^a^	1.002 (0.998–1.006)	1.001 (0.999–1.003)
eGFR, mL/min/1.73 m^2^	1.000 (0.993–1.008)	1.009 (1.000–1.018) ^a^	1.011 (0.999–1.023)	1.000 (0.992–1.008)
Uric acid, mg/dL	1.020 (0.929–1.120)	1.059 (0.955–1.173)	1.205 (0.973–1.492)	1.168 (1.018–1.341) ^a^
Calcium, mg/dL	1.541 (1.097–2.164) ^a^	1.384 (0.964–1.988)	1.314 (0.705–2.447)	1.361 (0.923–2.007)
CRP ≥ 75th percentile	1.006 (0.777–1.301)	1.247 (0.945–1.645)	1.483 (0.893–2.462)	1.233 (0.879–1.728)
Hypertension, yes vs. no	7.802 (5.819–10.461) ^c^	7.146 (5.225–9.773) ^c^	25.526 (12.032–54.152) ^c^	7.360 (4.776–11.343) ^c^
Diabetes, yes vs. no	0.853 (0.505–1.440)	2.730 (1.811–4.114) ^c^	2.394 (0.925–6.199)	1.009 (0.635–1.603)
KSD, yes vs. no	1.106 (0.777–1.574)	1.546 (1.111–2.151) ^a^	1.059 (0.496–2.259)	1.783 (1.042–3.054) ^a^
Smoking				
Ex vs. non	0.886 (0.593–1.322)	1.209 (0.796–1.838)	0.000 (0.000–0.000)	0.410 (0.035–4.840)
Current vs. non	1.295 (0.940–1.784)	1.622 (1.082–2.432) ^a^	2.333 (0.704–7.724)	0.913 (0.174–4.801)
Alcohol consumption				
Ex vs. non	0.697 (0.352–1.381)	0.710 (0.394–1.280)	0.000 (0.000–0.000)	1.078 (0.215–5.404)
Current vs. non	1.138 (0.829–1.563)	0.741 (0.519–1.057)	0.275 (0.056–1.340)	0.827 (0.280–2.440)
Regular exercise, yes vs. no	0.471 (0.294–0.752) ^b^	0.760 (0.517–1.116)	0.733 (0.316–1.698)	1.297 (0.737–2.283)

KSD: kidney stone disease; OR: odds ratio; CI: confidence interval; BMI: body mass index; HDL-C: high-density lipoprotein cholesterol; eGFR: estimated glomerular filtration rate, CRP: C-reactive protein. ^a^ Statistically significant, *p* < 0.05. ^b^ Statistically significant, *p* < 0.01. ^c^ Statistically significant, *p* < 0.001.

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
