# Peer review of "Relationship between Kidney Stone Disease and Arterial Stiffness in a Taiwanese Population"

_jcm, 2020, doi:10.3390/jcm9061693_

Round 1
Reviewer 1 Report
In this study, data from a large cohort of individuals who underwent health screening at a single centre in Taiwan were analysed to investigate a possible association of kidney stone disease with arterial stiffness. The association was confirmed for stones (asymptomatic) identified by abdominal ultrasound in men and women over 50 years of age, which was independent of the other risk factors for arterial stiffness analysed. Neither of the only two previous studies of the association addressed age and gender. This is an intriguing finding still wanting an explanation, which should be included in further considerations of the aetiology of kidney stones. The report is well-written
2.1 Subjects. Who were they? from the general population?; Why were they chosen for a rather extensive 'health package' which includes ultrasound?; was this a 'one-off' assessment or a regular event? if so are previous data available?
2.4 Measurement of arterial stiffness:
L. 105 Why was the result for the right leg only used-why not an average of both as in ref 16?
Do you have any data for imprecision of the measurements?
3.2 Results
Age is a critical conclusion from your findings and merits a more detailed presentation. The peak age for symptomatic stones world-wide (including East Asia) is 40-49 years; Why was 50 years chosen to segregate the stiffness data?
Histograms (men & women separately) showing on the same plot: age distributions for the entire cohort, for arterial stiffness and for stones might be informative. From the ORs, the frequency of stiffness seems to peak earlier than the frequency for stones. This would be relevant for your discussion of mechanisms.
Stones. There must be more data you could extract from the ultrasound scans? Bilateral stones might indicate a systemic (or genetic) cause; multiple stones suggest a significant stone problem. This information may be relevant to the discussion.
4. Discussion
Possible explanations for the observed association with stones is that it occurred by chance; that they have a common basis which was not detectable from the health screen tests (amply discussed); that stones increase the risk for stiffness (seems unlikely from the evidence); or that stiffness of the large systemic arteries in some way adds to the other risk factors that lead to stones (eg disordered mineral transport in the kidneys). The possible impact of increased flow rate in the large arteries on flow and permeability of the small renal medullary arcuate blood vessels perhaps merits a mention in view of their central role in mineral transport in the distal nephron
L.170 perhaps 'uknown' is more accurate than 'uncertain'
L.219-230 Too much on exercise which is not very relevant
Reviewer 2 Report
Authors compared kidney stone (KSD) and other variables including CVD risk factors and serum calcium content between those with/without arterial stiffness evaluated with baPWV, and calculated odds ratio or arterial stiffness associated with those risk factors and KSD. Therefore, the title of the manuscript “Relationship between kidney stone disease and arterial stiffness in a Taiwanese population” does not match for the content. Obesity, diabetes, hypertension and metabolic syndrome are considered risk factors for stone formation, which are common cause for arterial stiffening. Therefore, it is no surprise that more people with kidney stone are suffered from arterial stiffness and other CVD risk factors. What is the purpose of this study? What implication can authors derive from the current results?
INTRODUCTION
The first paragraph (cfPWV vs baPWV) is not necessary. Authors must describe clearly what is the clinical problem regarding kidney stone and arterial stiffness (research question). What merit could be achieved when the research question was answered?
METHODS
To reconcile the manuscript title and the statistics, statistics should have been done for “those with/without kidney stone”. And the variables may have been age, BMI, BP, cholesterol, ,,,, arterial stiffness. But, they are risk factors for both KSD and arterial stiffness. Funny.
Reviewer 3 Report
Evaluation of the review paper jcm-812201 entitled “Relationship between Kidney Stone Disease and Arterial Stiffness in a Taiwanese Population”. This paper has relevance to the purpose and the audience of jcm.
The aim of this paper was to investigate the associations between kidney stone disease (KSD) and increased arterial stiffness using multiple logistic regression models in 6,694 male and female subjects from October 2006 to August 2009.
The authors suggest that KSD is associated with a higher risk of increased arterial stiffness in individuals aged ≥ 31 50 years, but not in those aged < 50 years for both gender
Major Comments for the authors
- This paper is focused on an interesting issue: the association between KSD and increased arterial stiffness.
- The text and the tables are of appropriate length and informative.
-
The references are up to date.
- The results of the paper have practical implications for patients with KSD.
-
I believe that references 15 and 16, previous studies on the same issue, should be analysed and discussed more extensively.
15. Fabris A1, Ferraro PM2, Comellato G3, Caletti C1, Fantin F3, Zaza G1, Zamboni M3, Lupo A1, Gambaro G4. The relationship between calcium kidney stones, arterial stiffness and bone density: unraveling the stone-bone-vessel liaison. J Nephrol 2015 Oct;28(5):549-55.
16. Fan, X.; Kalim, S.; Ye, W.; Zhao, S.; Ma, J.; Nigwekar, S.U.; Chan, K.E.; Cui, J.; Cai, J.; Wang, L.; et al. Urinary 292 stone disease and cardiovascular disease risk in a rural chinese population. Kidney Int Rep 2017, 2, 1042-293 1049.
Round 2
Reviewer 2 Report
Authors added Table S1 showing characteristics of participants with or without KSD. The results were overwhelmingly similar to Table 1 showing characteristics of participants with or without arterial stiffness. The resemblance of Table S1 and Table 1 is so impressive, thus the results in Table 2 can be expected. Table S1 can be Table 2 (not supplemental table).
Figure 1 clearly shows that the relationship with age is different for arterial stiffness and KSD. Arterial stiffness increases in a linear fashion which may be same with BP or glucose tolerance. Prevalence of KSD dicrease for 70 years and over which might be related to decreased urinary concentrating capacity.
Author Response
Department of Family Medicine
National Cheng Kung University Hospital
May 28, 2020
Dear Reviewer:
Thank you very much for giving us the valuable comments for the revised manuscript, which was entitled "Relationship between Kidney Stone Disease and Arterial Stiffness in a Taiwanese Population". We hope our revision will satisfy your suggestions.
Yours sincerely,
Corresponding Author:
Jin-Shang Wu, MD, MS, Department of Family Medicine, National Cheng Kung University Hospital, No.138, Sheng Li Road, Tainan city 70403, Taiwan.
Tel: +886-6-2353535
Email: jins@mail.ncku.edu.tw
Co-corresponding Author:
Chih-Jen Chang, MD
E-mail address: changcj.ncku@gmail.com
Response to Reviewer 2 Comments
Comments and Suggestions for Authors
Comment #1
|
Authors added Table S1 showing characteristics of participants with or without KSD. The results were overwhelmingly similar to Table 1 showing characteristics of participants with or without arterial stiffness. The resemblance of Table S1 and Table 1 is so impressive, thus the results in Table 2 can be expected. Table S1 can be Table 2 (not supplemental table). |
Response: Thank you for your suggestion. We changed Table S1 into Table 2 and added it in RESULTS section of the revised manuscript.
The revised part of manuscript was as follows.
|
Section |
Original part |
Revised part |
|
RESULTS Page 3-5 of the revised manuscript |
Moreover, the comparisons of the same variables between subjects with and without KSD are also shown in Table S1 (Supplementary Materials). The results are similar with that of the comparisons between subjects with and without increased arterial stiffness except for calcium and CRP ≥75th percentile (Table S1). |
Moreover, the comparisons of the same variables between subjects with and without KSD are also shown in Table 2S1 (Supplementary Materials). The results are similar with that of the comparisons between subjects with and without increased arterial stiffness except for calcium and CRP ≥75th percentile (Table 2). |
Table 2. Comparisons of clinical characteristics between subjects with and without kidney stone disease (KSD).
|
|
KSD |
||
|
No (n=6034) |
Yes (n=660) |
p value |
|
|
Age, years |
47.2 ± 12.0 |
50.4 ± 11.1 |
<0.001 |
|
Male gender |
3517 (58.3) |
483 (73.2) |
<0.001 |
|
BMI, kg/m2 |
24.1 ± 3.5 |
24.6 ± 3.1 |
0.001 |
|
SBP, mmHg |
118.9 ± 16.2 |
123.8 ± 16.3 |
<0.001 |
|
DBP, mmHg |
70.6 ± 11.2 |
74.9 ± 11.4 |
<0.001 |
|
FPG, mg/dL |
89.3 ± 18.1 |
91.6 ± 20.2 |
0.003 |
|
2-h PG, mg/dL |
117.6 ± 49.9 |
123.7 ± 54.7 |
0.003 |
|
HbA1c, % |
5.67 ± 0.7 |
5.74 ± 0.7 |
0.006 |
|
Total cholesterol, mg/dL |
197.1 ± 37.2 |
200.6 ± 34.1 |
<0.001 |
|
Triglyceride, mg/dL |
125.8 ± 86.3 |
137.5 ± 97.7 |
<0.001 |
|
HDL-C, mg/dL |
51.7 ± 14.4 |
49.5 ± 13.6 |
<0.001 |
|
Total cholesterol/HDL-C |
4.1 ± 1.3 |
4.3 ± 1.2 |
<0.001 |
|
Creatinine, mg/dL |
0.86 ± 0.29 |
0.90 ± 0.19 |
0.002 |
|
eGFR, mL/min/1.73m2 |
94.7 ± 18.2 |
92.4 ± 18.0 |
0.002 |
|
Uric acid, mg/dL |
5.8 ± 1.5 |
6.1 ± 1.6 |
<0.001 |
|
Calcium, mg/dL |
9.2 ± 0.4 |
9.2 ± 0.4 |
0.813 |
|
CRP ≥75th percentile |
1484 (24.6) |
180 (27.3) |
0.131 |
|
Hypertension |
979 (16.2) |
148 (22.4) |
<0.001 |
|
Diabetes |
422 (7.0) |
65 (9.8) |
0.007 |
|
Increased arterial stiffness |
1973 (32.7) |
307 (46.5) |
<0.001 |
|
Smoking |
|
|
0.01 |
|
Non |
5085 (84.3) |
541 (82.0) |
|
|
Ex |
374 (6.2) |
61 (9.2) |
|
|
Current |
575 (9.5) |
58 (8.8) |
|
|
Alcohol consumption |
|
|
0.218 |
|
Non |
5214 (86.4) |
1956 (84.7) |
|
|
Ex |
154 (2.6) |
65 (3.6) |
|
|
Current |
666 (11.0) |
259 (11.7) |
|
|
Regular exercise |
448 (7.4) |
53 (8.0) |
0.574 |
|
Data are presented as mean ± SD or number (%). BMI: body mass index; SBP: systolic blood pressure; DBP: diastolic blood pressure; HbA1c: glycated hemoglobin; FPG: fasting plasma glucose; 2-h PG: two-hour post-load glucose; HDL-C: high-density lipoprotein cholesterol; eGFR: estimated glomerular filtration rate; CRP: C-reactive protein; KSD: kidney stone disease. The Student’s t-test for continuous variables or the Chi-square test for categorical variables. |
|||
Comment #2
|
Figure 1 clearly shows that the relationship with age is different for arterial stiffness and KSD. Arterial stiffness increases in a linear fashion which may be same with BP or glucose tolerance. Prevalence of KSD decrease for 70 years and over which might be related to decreased urinary concentrating capacity. |
Response: Thank you for your comment. We have revised the description for the relationship between age and KSD as follows. Additional references were cited and added to REFERENCES section.
The revised part of manuscript was as follows.
|
Section |
Original part |
Revised part |
|
DISCUSSION page 8 of the revised manuscript |
The results in this study concurred with previous findings. Exercise has anti-oxidative and anti-inflammatory effects and can enhance the production of NO and reduce the concentrations of vasoconstrictors, …… |
The results in this study concurred with previous findings. Similar to arterial stiffness, the prevalence of KSD also showed a significant increasing trend with advancing age, but it declined in the age group ≥ 70 years in total subjects. The explanation for this may be that aged people have reduced urinary concentrating ability in response to vasopressin [44], which may associate with hypercalciuria, the most common metabolic abnormality in calcium stone formers [45]. Exercise has anti-oxidative and anti-inflammatory effects and can enhance the production of NO and reduce the concentrations of vasoconstrictors, …… |
|
REFERENCES page 11 of the revised manuscript |
- |
44. Tamma, G.; Goswami, N.; Reichmuth, J.; De Santo, N.G.; Valenti, G. Aquaporins, Vasopressin, and Aging: Current Perspectives. Endocrinology 2015, 156, 777-788. 45. Procino, G.; Mastrofrancesco, L.; Tamma, G.; Lasorsa, D.R.; Ranieri, M.; Stringini, G.; Emma, F.; Svelto, M.; Valenti, G. Calcium-sensing receptor and aquaporin 2 interplay in hypercalciuria-associated renal concentrating defect in humans. An in vivo and in vitro study. PloS one 2012, 7, e33145. |
